# Association between the Portion Sizes of Traditional Japanese Seasonings—Soy Sauce and Miso—and Blood Pressure: Cross-Sectional Study Using National Health and Nutrition Survey, 2012–2016 Data

**DOI:** 10.3390/nu10121865

**Published:** 2018-12-02

**Authors:** Emiko Okada, Aki Saito, Hidemi Takimoto

**Affiliations:** 1Department of Nutritional Epidemiology, National Institute of Biomedical Innovation, Health and Nutrition, 1-23-1 Toyama Shinjuku-ku, Tokyo 162-8636, Japan; okadae@nibiohn.go.jp (E.O.), saitoa@nibiohn.go.jp (A.S.); 2Office of Nutrition, Health Service Division, Health Service Bureau, Ministry of Health, Labour and Welfare, 1-2-2 Kasumigaseki Chiyoda-ku, Tokyo 100-8916, Japan

**Keywords:** Japanese seasoning, blood pressure, hypertension, National Health and Nutrition Survey, Japan

## Abstract

While previous studies have reported the association between food portion size and health outcomes, the association between food seasonings and blood pressure (BP) has not been clarified. This study aimed to investigate the association between the portion sizes of traditional Japanese seasonings and BP. Data on 25,738 Japanese participants (10,154 men and 15,584 women) aged ≥20 years registered in the 2012–2016 National Health and Nutrition Survey (NHNS) were used for this cross-sectional study. The portion sizes of soy sauce or miso were calculated from the reported weight of soy sauce or miso, and the number of dishes. Regression models were used to estimate BP and prevalence of hypertension, and their 95% confidence intervals according to the portion size of soy sauce or miso. We did not observe an association between the portion sizes of soy sauce or miso and BP. A larger portion size of soy sauce or miso was associated with a higher prevalence of hypertension in the crude model among women, but no significant associations were observed in the multivariate model in both sexes. Our findings suggest that the portion sizes of soy sauce or miso are not associated with BP among Japanese adults.

## 1. Introduction

Elevated blood pressure (BP) is one of the leading risk factors for mortality, and is estimated to have caused 9.4 million deaths and contributed to 7% of the disease burden worldwide, as measured in disability-adjusted life-years in 2010 [1]. The global prevalence of hypertension (defined as a systolic and/or diastolic BP ≥140/90 mmHg) in adults aged 18 years and older was estimated to be around 22% in 2014 [2]. In Japan, the number of patients with hypertensive disease who continuously received medical treatment was estimated to be 10,108,000 according to the 2014 Patient Survey [3]; this value accounts for nearly 8% of the total population. Some of the known modifiable risk factors for hypertension are overweight, obesity, high salt intake, insufficient intake of fruits and vegetables, high alcohol consumption, and physical inactivity [2]. Japanese people have higher salt intakes than those in Western countries [4]. A previous study that represented the Japanese population observed a positive relationship between dietary salt intake and BP levels [5]. Furthermore, a higher sodium intake was found to be associated with a higher risk of cardiovascular disease (CVD) in the Japanese population [6,7]. Seasonings are major dietary sources of salt among Japanese people [8]. In particular, traditional Japanese seasonings such as *shoyu* (soy sauce) and miso (fermented soy bean paste) have a higher sodium content than other seasonings; for example, *koikuchi-shoyu* (common type) contains 5,700 mg of sodium per 100 g, and rice-*koji* miso (light yellow type) contains 4,900 mg of sodium per 100 g [9]. Sodium intakes from soy sauce and miso soup account for 20% and 9.7%, respectively, of the total sodium intake among Japanese people [8].

It is important to consider food portion sizes, which mean the amount of food consumed per dish, to evaluate the effect of food intake on human health outcomes. Several studies have reported that the intake of larger portions of food with a high-energy density was associated with a high body mass index (BMI) or overweight in the general population [10,11]. However, it is not clear whether the portion sizes of seasonings are associated with health outcomes. The portion sizes of foods with a high sodium content are of importance in the prevention of hypertension. Traditional Japanese seasonings, especially soy sauce, are used not only in cooking, but are also added directly to dishes. Therefore, it is necessary to estimate the amount of traditional Japanese seasonings per dish per person, and investigate their association with BP. Since the Japan National Health and Nutrition Survey (NHNS) applies a dietary record method, it is possible to calculate the food portion size of individuals in detail.

Here, we aimed to explore the association between the portion sizes of traditional Japanese seasonings—soy sauce and miso—and BP levels among Japanese adults using nationwide data from the 2012–2016 NHNS.

## 2. Materials and Methods

### 2.1. National Health and Nutrition Survey Data

The NHNS is a cross-sectional annual survey that has been conducted by the Ministry of Health, Labour and Welfare in Japan between October and December since 1947. Details of the survey design have been described elsewhere [12]. Briefly, the NHNS uses a stratified single-stage cluster sample design, and is conducted across 47 prefectures (Kumamoto prefecture was excluded from the 2016 survey due to the earthquakes in that area). Census enumeration areas were drawn from each prefecture and residents aged ≥1 year in all households from 300 selected census enumeration areas in the 2013–2015 survey, 475 areas in the 2012 survey, and 462 areas in the 2016 survey were eligible for participation in the survey. The NHNS comprises three surveys: the dietary intake survey (a self-administered questionnaire including questions on the household population, meal patterns, daily step counts, and dietary records); the lifestyle survey (a self-administered questionnaire including questions on smoking status, alcohol intake, and sleep time); and a physical examination (measurement of height, weight, abdominal circumference, and BP; blood tests; and a medical interview). Based on official application procedures under Article 33 of the Statistics Act, we obtained approval from the Ministry of Health, Labour and Welfare, Japan and used individual-level data from the NHNS for this study. In accordance with the Ethical Guidelines of Epidemiological Research [13], our study was exempt from the application of these guidelines as only anonymized data were used.

### 2.2. Dietary Assessment

The dietary intake survey, which uses semi-weighed household dietary records to assess dietary intake, was conducted on a single day between October and December, excluding Sundays and public holidays. Trained interviewers visited each household to explain the method of generating dietary records before the survey. Dietary records were weighed by taking an inventory of all the foods and beverages consumed, food waste, leftovers, and foods consumed away from home in the household. For shared dishes within the household, the approximate proportions of each food were assigned to individual household members for the estimation of the individual food intakes. Interviewers checked for any missing information and errors during household visits to collect dietary records.

Nutrient intakes were calculated based on the Standard Tables of Food Composition in Japan, 2010, which were updated from the fifth revised and enlarged edition published in 2005. Foods were classified into 17 large food groups (e.g., cereals, vegetables, and fish and shellfish, etc.), 33 medium groups (e.g., rice and rice products, wheat flour and wheat products, green and yellow vegetables, other vegetables, raw fish and shellfish, and seafood and processed products), and 98 small groups (e.g., rice, bread, tomatoes, carrots, horse mackerels and sardines, and salmon and trout) based on the food group tables in the NHNS [14]. Details on the food groups have been previously reported [15].

### 2.3. Portion Size

For each participant, the portions sizes of soy sauce and miso were calculated by dividing the total weight of the soy sauce and/or miso consumed in all the meals of the day by the number of dishes that used soy sauce and/or miso. For example, if one participant consumed broiled fish with 5 g of soy sauce at breakfast, teriyaki chicken with 10 g of soy sauce at lunch, and *nikujaga* (simmered meat and potatoes) with 10 g of soy sauce and *hourensou-ohitashi* (boiled spinach) with 1 g of soy sauce at dinner, the portion size of soy sauce was calculated as 6.5 g (26 g/4 dishes). The total consumption of soy sauce included the following types of soy sauce: *koikuchi-shoyu* (common type, food number 17007), *usukuchi-shoyu* (light color type, 17008), *tamari-shoyu* (full-bodied type, 17009), *saishikomi-shoyu* (refermented type, 17010), *shiro-shoyu* (extra light color type, 17011), and low-salt *shoyu* (19101). The total consumption of miso included the consumption of the following types: rice-*koji* miso (sweet type, 17044), rice-*koji* miso (light yellow type, 17045), rice-*koji* miso (dark yellow type, 17046), barley-*koji* miso (17047), soybean-*koji* miso (17048), and low-salt miso (19102). For statistical analyses, participants other than those with an intake of 0 g of soy sauce/miso were divided into tertiles based on their portion sizes of soy sauce/miso by sex. Finally, participants were classified according to their portion sizes of soy sauce or miso into the following four groups: soy sauce (0 g, 0–<4.0 g, 4.0–<7.23 g, ≥7.23 g for men, 0 g, 0–<3.5 g, 3.5–<6.5 g, ≥6.5 g for women), miso (0 g, 0–<9.0 g, 9.0–<13.3 g, ≥13.3 g for men, 0 g, 0–<8.67 g, 8.67–<12.0 g, ≥12.0 g for women).

### 2.4. Blood Pressure Measurement

BP was measured twice using the Riva–Rocci mercurial sphygmomanometer and the JIS manchette (BP cuff). The mercury sphygmomanometer will be replaced with an alternative device no later than 2020, and then it will go out of production, according to the World Health Organization recommendation. Details on the blood pressure measurement have been previously reported [15]. Briefly, participants were in a resting state for 5 minutes or longer before the measurement. The mercurial sphygmomanometer was placed in a vertical position. BP was measured with the right upper arm in a sitting position. The rubber bladder of the manchette was placed around the right upper arm of the participants to cover the brachial artery. The center of the manchette was on the same level as the participant’s heart. Systolic BP (SBP) was estimated by the palpation method, then the pressure in the manchette was dropped to zero (0) for a short while by deflating it. After raising the pressure again to a level more than 30 mmHg higher than that estimated by the palpation method, the SBP and diastolic BP (DBP) were finally measured with the aid of a stethoscope. After a one or two minute interval, the manchette was placed around the participant’s arm again, and then the BP was measured for the second time. 

For the statistical analyses, the mean values of the two measurements of SBP and DBP were used. Hypertension was defined as an SBP ≥140 mmHg and/or a DBP ≥90 mmHg in this study based on the 2014 Guidelines for the Management of Hypertension by the Japanese Society of Hypertension [16].

### 2.5. Study Participants

Of the 93,577 participants in the 2012–2016 survey, participants aged <20 years (*n* = 14,533), pregnant or breastfeeding women (*n* = 874), those in whom BP was not measured (*n* = 40,644), and those who did not complete the dietary intake survey (*n* = 1,126) were excluded. Furthermore, 10,662 participants who answered “using drugs to lower BP” in the medical interview were excluded. Finally, a total of 25,738 participants (10,154 men and 15,584 women) aged ≥20 years were included in the analysis. We analyzed the results of the present study using the NHNS as a simple random sample.

### 2.6. Statistical Analysis

The associations between the portion sizes of soy sauce and miso and participant characteristics and food and nutrient intakes were assessed using simple linear regression analyses for continuous variables, and a chi-square test for categorical variables. Age-adjusted model and multivariate model liner regression analyses were performed to evaluate the associations between the portion sizes of soy sauce or miso and SBP or DBP. For the association between the portion sizes of soy sauce or miso and hypertension, age-adjusted model and multivariate model logistic regression analyses were performed. The odds ratios (ORs) for hypertension were evaluated using the portion sizes of soy sauce or miso in the second to fourth groups, and these were compared to those in the first group. To examine the linear relationship, *P-*for-trend values were obtained using the portion sizes of soy sauce or miso as continuous variables. The following variables were included in the multivariate models based on a review of the previous literature: age (continuous variable), BMI (<18.5, 18.5–24.9, 25.0–29.9, ≥30.0 kg/m^2^, or unknown), amount of alcoholic beverages intake (0, 0–<270, ≥ 270 g/day for men; 0, 0–<9.0, ≥9.0 g/day for women), number of steps (<4896, 4896–8242, ≥8243/day for men, <4605, 4605–7480, ≥7481/day for women), and consumption of vegetables (g/day, continuous variable) and fruits (g/day, continuous variable). In addition, we performed a multivariate logistic regression analyses for high normal blood pressure, defined as an SBP 130–139 mmHg and/or a DBP 85–89 mmHg, and normal blood pressure defined as an SBP 120–129 mmHg, and/or a DBP 80–84 mmHg [16]. All statistical analyses were performed using the SAS statistical package for Windows (version 9.4, SAS Institute Inc., Cary, NC, USA). Differences were considered to be statistically significant at *p* < 0.05.

## 3. Results

The median portion size values were 4.0 g for men and 3.5 g for women (soy sauce), and 9.0 g for men and 8.67 g for women (miso). The demographic characteristics of the participants in terms of the portion sizes of the traditional Japanese seasonings by the four groups based on sex are shown in Table 1 (soy sauce) and Table 2 (miso). Participants with larger portion sizes of soy sauce were older and likelier to consume alcohol, and the women in this group were less likely to be current smokers. Participants with larger portion sizes of miso were also older and less likely to be current smokers.

The characteristics of the dietary intakes of the participants in terms of the portion sizes of the traditional Japanese seasonings by the four groups based on sex are shown in Table 3 (soy sauce) and Table 4 (miso). Participants with a larger portion size of soy sauce had a higher intake of almost all foods and nutrients, but a lower intake of milk and fat (%energy). Participants with a larger portion size of miso also had a higher intake of almost all foods and nutrients, but a lower intake of meat, milk, fats and oils (g), confectioneries, and fat (%energy).

Table 5 shows the adjusted-mean BP levels according to the portion sizes of traditional Japanese seasonings by the four groups based on sex. No associations between the portion sizes of soy sauce and SBP or DBP level were observed in the crude and multivariate models among men. In women, a larger portion size of soy sauce was associated with a higher SBP level in the crude and age-adjusted model (*p* for trend = 0.026), but not in the multivariate model. For miso, larger portion sizes were associated with higher SBP or DBP levels in the crude model in both men and women, but not in the age-adjusted or multivariate models.

The results of the multivariate logistic regression analyses for the ORs of hypertension according to the four groups of traditional Japanese seasoning portion size by sex are shown in Table 6. A total of 7,279 of the participants had hypertension. A larger portion size of soy sauce was associated with a higher prevalence of hypertension among women, with a crude OR of 1.25 (95% confidence interval (CI): 1.11, 1.41) in the fourth group (*p* for trend < 0.001). However, in both the age-adjusted and multivariate models, no significant association between the portion size of soy sauce and hypertension were observed in both sexes. As for miso, a larger portion size was also associated with a higher prevalence of hypertension among women in the crude model (OR in the fourth group = 1.37 (95% CI: 1.24, 1.51); *p* for trend < 0.001), but in the age-adjusted and multivariate model, no significant associations were observed in both sexes. Guidelines for hypertension were revised by the American College of Cardiology/American Heart Association in 2017 [17]. We performed an additional analysis for high normal blood pressure and normal blood pressure based on the 2014 Guidelines for the Management of Hypertension in Japan [16]. However, no significant association was observed in the multivariate model [18].

## 4. Discussion

We investigated the association between the portion sizes of traditional Japanese seasonings—soy sauce and miso—and BP levels in the general Japanese population, using data from the nationwide NHNS. The portion sizes of soy sauce or miso were not associated with BP levels and the prevalence of hypertension among Japanese adults. The median portion size values of traditional Japanese seasonings correspond to about one teaspoon of soy sauce and about two teaspoons of miso using the teaspoon used for cooking. Soy sauce (4.0 g) contains 228 mg of sodium (*koikuchi-shoyu*), and miso (9.0 g) contains 441 mg of sodium (rice-koji miso) [9].Therefore, the portion sizes of Japanese seasonings calculated in this study are reasonable, considering the dietary habits of Japanese people [19]. Since participants generally find it difficult to record the precise amount of seasoning for one day, it is beneficial to assess the amount using portion sizes, which correspond to the food intake per dish.

It is well-known that a high salt intake is associated with increased BP. In Japanese people too, a high dietary salt intake was associated with high BP levels [5], as observed in a previous study using NHNS data. However, another study indicated that the frequency of miso soup consumption was not associated with hypertension among Japanese people [20]. In our study, the portion sizes of soy sauce and miso were associated with higher BP levels and the prevalence of hypertension in the crude model among women. However, after adjusting for age and confounders, including dietary factors affecting BP, such as vegetable and fruit intake, BP was not affected by the portion sizes of soy sauce or miso. Conventionally, in Japan, when people consume soy sauce or miso, they also consume vegetables that are reported to lower BP [21,22]. Furthermore, participants with larger portion sizes of soy sauce or miso had a higher intake of potassium contained in vegetables, fruits, and fish, as well as sodium in this study. As soy sauce and miso are produced from soy and wheat, they also contain a large amount of potassium (390 mg per 100 g in soy sauce, 440 mg per 100 g in miso) [9]. A high intake of potassium reduces BP and lowers the risk of stroke, as observed in a meta-analysis including 35 randomized controlled trials and cohort studies [23]. The demographic and dietary characteristics of participants with larger portion sizes of soy sauce or miso correspond to older people and those with a higher intake of almost all nutrients and foods, including vegetables, fruits, fish, and soy products; this may offset the effects of soy sauce or miso on BP. Some cohort studies in Japan have shown that the dietary pattern characterized by a high consumption of vegetables, fish, soy products, and seaweed was associated with a reduced risk of CVD [24,25]. Therefore, the larger portion size of soy sauce or miso may not have been associated with hypertension, according to traditional Japanese dietary patterns in our study.

Apart from seasonings, dietary salt is also obtained from other foods and processed foods. The International Study of Macro-and Micro-Nutrients showed that, in the United States and United Kingdom, breads, grains, cereals, red meats, poultry, eggs, and dairy products are the main dietary sources of sodium, while in Japan, soy sauce, Japanese pickles, and miso soup are the main dietary sources of sodium [8]. We previously reported that, based on NHNS data, the major dietary salt sources among Japanese people are Chinese noodles (dried by frying), Japanese curry roux, Japanese pickles (pickled in salty vegetables), and *umeboshi* (Japanese apricots; pickled and dried mume), besides miso and soy sauce [26]. Soy sauce and miso are major dietary sources of salt, but their portion sizes may not directly affect BP. Recently, low sodium seasonings have become popular in the marketplace. A previous Japanese study has shown that use of low-sodium soy sauce and miso were not effective in lowering BP after the 6-week intervention, but that DBP decreased in those 40 years and older [27]. Because the present study could not investigate the portion size of low sodium soy sauce and miso, and because our study design differed from this previous study, effects of low sodium seasonings on BP are not clear in our study. The latest version of the Standard Tables of Food Composition has provided new food codes for low-sodium seasonings [9], so the prevalence of usage can be monitored in the future NHNS.

Soy sauce and miso are fermented foods produced using soybeans as a raw material. Higher intakes of soy sauce and miso may be associated with lower concentrations of the inflammatory marker interleukin-6 among Japanese people [28]. Moreover, the soy isoflavones contained in soybeans are indicated to have a BP-lowering effect in people with hypertension, as observed in a meta-analysis of randomized controlled trials [29]. A recent prospective study showed that the intake of fermented soy products was inversely associated with high BP development in the general Japanese population with a normal BP [30]. Therefore, larger portion sizes of soy sauce and miso may not be associated with elevated BP levels for these reasons.

The present study has some limitations. First, the association derived from this cross-sectional study could reverse the causal association between the traditional Japanese seasonings and BP levels. However, we excluded participants using BP-lowering drugs from this analysis. Second, we calculated the portion sizes of soy sauce or miso from the dietary records of households with proportional distributions within the house. Reliance on household representatives for the recording of dietary intakes in the survey may have resulted in the misreporting of various foods, since Japanese workers typically eat lunch outside the house during the week. A previous study investigated the validity of determining the food consumption of individual family members through household-based and individual-based food weighing methods, and showed that the total energy and macronutrient consumptions of individual participants displayed a high level of agreement among the household-based and individual-based food weighing methods [31]. Therefore, using dietary records from households with proportional distributions is a valid method for the estimation of individual intakes in the NHNS. Third, dietary records may not reflect average individual dietary habits, as this survey used dietary records from a single weekday. Additionally, individual habitual dietary intakes naturally vary between weekdays and weekends. However, our previous study using 2012 NHNS data showed that food intake patterns derived from this one-day survey were related to cardiovascular risk factors, such as waist circumference, BMI, SBP, DBP, and blood lipid profiles [32]. Therefore, we speculate that the current dietary survey method can minimize day-to-day variations and reflect habitual intakes at the population level. Fourth, estimated food and nutrient intakes may not be the true values. Although the trained interviewers explained the method of generating dietary records before the survey and checked for any missing information and errors during household visits, dietary intake values may have been under- or over-reported, because the NHNS uses self-administered dietary records. Finally, because there were participants who did not participate in the physical examinations, and BMI was unavailable for some participants, it is possible that the BMI distribution among groups for traditional Japanese seasonings may be unbalanced/unmatched.

## 5. Conclusions

In conclusion, when adjusting for appropriate risk factors, the portion sizes of soy sauce and miso were not associated with BP levels and the prevalence of hypertension among Japanese adults. Our findings suggest that larger portion sizes of soy sauce or miso alone may not be risk factors for elevated BP. In fact, traditional diets using soy sauce or miso as seasonings may produce health benefits, through the higher intakes of vegetables and soy products. Further prospective studies are required to examine the association between traditional Japanese seasonings and BP levels.

## Figures and Tables

**Table 1 nutrients-10-01865-t001:** Demographic characteristics of the participants according to the portion size of soy sauce.

		Men									Women								
		Group 1 (0 g)		Group 2 (0–<4.0 g)		Group 3 (4.0–<7.23 g)		Group 4 (≥7.23 g)		*p* Value	Group 1 (0 g)		Group 2 (0–<3.5 g)		Group 3 (3.5–<6.5 g)		Group 4 (≥6.5 g)		*p* Value
		(*n* = 1350)		(*n* = 2982)		(*n* = 2826)		(*n* = 2996)			(*n* = 2653)		(*n* = 4338)		(*n* = 4249)		(*n* = 4344)		
		*n*	(%)	*n*	(%)	*n*	(%)	*n*	(%)		*n*	(%)	*n*	(%)	*n*	(%)	*n*	(%)	
Age, (years) ^a^		54.2	(16.7)	55.8	(16.7)	57.2	(16.3)	55.9	(16.5)	<0.001	52.3	(15.3)	54.6	(15.7)	55.9	(15.6)	56.0	(15.1)	<0.001
BMI (kg/m^2^)	<18.5	38	(2.8)	79	(2.7)	79	(2.8)	80	(2.7)	0.008	184	(6.9)	290	(6.7)	266	(6.3)	284	(6.5)	0.041
	18.5–24.9	526	(39.0)	1163	(39.0)	1182	(41.8)	1199	(40.0)		1107	(41.7)	1830	(42.2)	1830	(43.1)	1876	(43.2)	
	25.0–29.9	185	(13.7)	417	(14.0)	409	(14.5)	461	(15.4)		200	(7.5)	351	(8.1)	354	(8.3)	414	(9.5)	
	≥30.0	32	(2.4)	55	(1.8)	43	(1.5)	86	(2.9)		52	(2.0)	58	(1.3)	69	(1.6)	78	(1.8)	
	Unknown	569	(42.2)	1268	(42.5)	1113	(39.4)	1170	(39.1)		1110	(41.8)	1809	(41.7)	1730	(40.7)	1692	(39.0)	
Smoking status	Non-smoker	748	(55.4)	1727	(57.9)	1617	(57.2)	1619	(54.0)	0.012	1584	(59.7)	2673	(61.6)	2443	(57.5)	2436	(56.1)	<0.001
	Smoker	456	(33.8)	906	(30.4)	852	(30.2)	992	(33.1)		292	(11.0)	282	(6.5)	316	(7.4)	339	(7.8)	
	Unknown	146	(10.8)	349	(11.7)	357	(12.6)	385	(12.9)		777	(29.3)	1383	(31.9)	1490	(35.1)	1569	(36.1)	
Alcohol drinking status	Current drinker	756	(56.0)	1758	(59.0)	1701	(60.2)	1774	(59.2)	0.041	859	(32.4)	1406	(32.4)	1365	(32.1)	1362	(31.4)	<0.001
	Former drinker	40	(3.0)	84	(2.8)	68	(2.4)	80	(2.7)		39	(1.5)	50	(1.2)	41	(1.0)	39	(0.9)	
	Never drinker	157	(11.6)	312	(10.5)	348	(12.3)	362	(12.1)		589	(22.2)	999	(23.0)	1108	(26.1)	1157	(26.6)	
	Unknown	397	(29.4)	828	(27.8)	709	(25.1)	780	(26.0)		1166	(44.0)	1883	(43.4)	1735	(40.8)	1786	(41.1)	
Amount of alcoholic beverages intake (g/day)	Q1	799	(59.2)	1199	(40.2)	999	(35.4)	1033	(34.5)	<0.001	2024	(76.3)	2407	(55.5)	2060	(48.5)	2026	(46.6)	<0.001
	Q2	204	(15.1)	922	(30.9)	933	(33.0)	1029	(34.4)		190	(7.2)	1180	(27.2)	1233	(29.0)	989	(22.8)	
	Q3	347	(25.7)	861	(28.9)	894	(31.6)	934	(31.2)		439	(16.6)	751	(17.3)	956	(22.5)	1329	(30.6)	
Exercise habits	No	539	(39.9)	1155	(38.7)	1140	(40.3)	1309	(43.7)	<0.001	1162	(43.8)	1914	(44.1)	1898	(44.7)	1974	(45.4)	0.007
	Yes	579	(42.9)	1296	(43.5)	1286	(45.5)	1241	(41.4)		966	(36.4)	1675	(38.6)	1625	(38.2)	1676	(38.6)	
	Unknown	232	(17.2)	531	(17.8)	400	(14.2)	446	(14.9)		525	(19.8)	749	(17.3)	726	(17.1)	694	(16.0)	
Number of steps (/day)	Q1	453	(33.6)	923	(31.0)	878	(31.1)	952	(31.8)	0.058	866	(32.6)	1317	(30.4)	1347	(31.7)	1411	(32.5)	0.028
	Q2	430	(31.9)	927	(31.1)	919	(32.5)	930	(31.0)		834	(31.4)	1364	(31.4)	1319	(31.0)	1423	(32.8)	
	Q3	393	(29.1)	1016	(34.1)	911	(32.2)	982	(32.8)		833	(31.4)	1482	(34.2)	1426	(33.6)	1348	(31.0)	
	Unknown	74	(5.5)	116	(3.9)	118	(4.2)	132	(4.4)		120	(4.5)	175	(4.0)	157	(3.7)	162	(3.7)	

^a^ Means (standard deviation). Amount of alcoholic beverages intake: Q1: 0 g/day, Q2: 0–<270 g/day (men), 0–<9.0 g/day (women), Q3: ≥ 270 g/day (men); ≥9.0 g/day (women). Steps: Q1: <4896/day (men), <4605/day (women), Q2: 4896–<8243/day (men), 4605–<7481/day (women), Q3: ≥8243/day (men), ≥7481/day (women). Simple linear regression analyses and a chi-square test were used for continuous and categorical variables, respectively. BMI, body mass index.

**Table 2 nutrients-10-01865-t002:** Demographic characteristics of the participants according to the portion size of miso.

		Men									Women								
		Group 1 (0 g)		Group 2 (0–<9.0 g)		Group 3 (9.0–<13.3 g)		Group 4 (≥13.3 g)		*p* Value	Group 1 (0 g)		Group 2 (0–<8.67 g)		Group 3 (8.67–<12.0 g)		Group 4 (≥12.0 g)		*p* Value
		(*n* = 3615)		(*n* = 2410)		(*n* = 1932)		(*n* = 2197)			(*n* = 5404)		(*n* = 3365)		(*n* = 3384)		(*n* = 3431)		
		*n*	(%)	*n*	(%)	*n*	(%)	*n*	(%)		*n*	(%)	*n*	(%)	*n*	(%)	*n*	(%)	
Age, (years) ^a^		52.6	(16.4)	58.5	(16.5)	57.2	(16.3)	57.6	(16.2)	<0.001	51.5	(15.4)	56.8	(15.6)	56.5	(15.2)	57.1	(14.9)	<0.001
BMI (kg/m^2^)	<18.5	100	(2.8)	72	(3.0)	44	(2.3)	60	(2.7)	<0.001	345	(6.4)	226	(6.7)	244	(7.2)	209	(6.1)	<0.001
	18.5–24.9	1375	(38.0)	989	(41.0)	797	(41.3)	909	(41.4)		2244	(41.5)	1406	(41.8)	1456	(43.0)	1537	(44.8)	
	25.0–29.9	534	(14.8)	290	(12.0)	291	(15.1)	357	(16.3)		436	(8.1)	273	(8.1)	290	(8.6)	320	(9.3)	
	≥30.0	93	(2.6)	42	(1.7)	33	(1.7)	48	(2.2)		111	(2.1)	41	(1.2)	45	(1.3)	60	(1.8)	
	Unknown	1513	(41.9)	1017	(42.2)	767	(39.7)	823	(37.5)		2268	(42.0)	1419	(42.2)	1349	(39.9)	1305	(38.0)	
Smoking status	Non-smoker	1907	(52.8)	1442	(59.8)	1104	(57.1)	1258	(57.3)	<0.001	3223	(59.6)	2017	(59.9)	1975	(58.4)	1921	(56.0)	<0.001
	Smoker	1312	(36.3)	670	(27.8)	586	(30.3)	638	(29.0)		578	(10.7)	192	(5.7)	220	(6.5)	239	(7.0)	
	Unknown	396	(11.0)	298	(12.4)	242	(12.5)	301	(13.7)		1603	(29.7)	1156	(34.4)	1189	(35.1)	1271	(37.0)	
Alcohol drinking status	Current drinker	2046	(56.6)	1455	(60.4)	1137	(58.9)	1351	(61.5)	0.008	1854	(34.3)	1043	(31.0)	1064	(31.4)	1031	(30.1)	<0.001
	Former drinker	103	(2.9)	75	(3.1)	51	(2.6)	43	(2.0)		67	(1.2)	34	(1.0)	21	(0.6)	47	(1.4)	
	Never drinker	438	(12.1)	269	(11.2)	234	(12.1)	238	(10.8)		1203	(22.3)	821	(24.4)	888	(26.2)	941	(27.4)	
	Unknown	1028	(28.4)	611	(25.4)	510	(26.4)	565	(25.7)		2280	(42.2)	1467	(43.6)	1411	(41.7)	1412	(41.2)	
Amount of alcoholic beverages intake (g/day)	Q1	1652	(45.7)	840	(34.9)	724	(37.5)	814	(37.1)	<0.001	3186	(59.0)	1666	(49.5)	1801	(53.2)	1864	(54.3)	<0.001
	Q2	901	(24.9)	867	(36.0)	617	(31.9)	703	(32.0)		1019	(18.9)	956	(28.4)	881	(26.0)	736	(21.5)	
	Q3	1062	(29.4)	703	(29.2)	591	(30.6)	680	(31.0)		1199	(22.2)	743	(22.1)	702	(20.7)	831	(24.2)	
Exercise habits	No	1515	(41.9)	913	(37.9)	792	(41.0)	923	(42.0)	<0.001	2469	(45.7)	1431	(42.5)	1467	(43.4)	1581	(46.1)	<0.001
	Yes	1488	(41.2)	1084	(45.0)	852	(44.1)	978	(44.5)		1916	(35.5)	1334	(39.6)	1360	(40.2)	1332	(38.8)	
	Unknown	612	(16.9)	413	(17.1)	288	(14.9)	296	(13.5)		1019	(18.9)	600	(17.8)	557	(16.5)	518	(15.1)	
Number of steps (/day)	Q1	1140	(31.5)	786	(32.6)	593	(30.7)	687	(31.3)	0.170	1716	(31.8)	1095	(32.5)	1038	(30.7)	1092	(31.8)	0.135
	Q2	1133	(31.3)	770	(32.0)	617	(31.9)	686	(31.2)		1668	(30.9)	1054	(31.3)	1119	(33.1)	1099	(32.0)	
	Q3	1162	(32.1)	769	(31.9)	651	(33.7)	720	(32.8)		1780	(32.9)	1096	(32.6)	1088	(32.2)	1125	(32.8)	
	Unknown	180	(5.0)	85	(3.5)	71	(3.7)	104	(4.7)		240	(4.4)	120	(3.6)	139	(4.1)	115	(3.4)	

^a^ Means (standard deviation). Amount of alcoholic beverages intake: Q1 (0 for men and women), Q2 (0–< 270 for men, 0–< 9.0 for women), and Q3 (≥ 270 for men, ≥ 9.0 for women). Steps: Q1 (< 4896 for men, < 4605 for women), Q2 (4896–8242 for men, 4605–7480 for women), and Q3 (≥ 8243 for men, ≥ 7481 for women). Simple linear regression analyses and a chi-square test were used for continuous and categorical variables, respectively. BMI, body mass index.

**Table 3 nutrients-10-01865-t003:** Characteristics of the dietary intakes of the participants according to the portion size of soy sauce.

	Men								Women							
	Group 1 (0 g)		Group 2 (0–<4.0 g)		Group 3 (4.0–<7.23 g)		Group 4 (≥7.23 g) ^a^		Group 1 (0 g)		Group 2 (0–<3.5 g)		Group 3 (3.5–<6.5 g)		Group 4 (≥6.5 g) ^a^	
	(*n* = 1350)		(*n* = 2982)		(*n* = 2826)		(*n* = 2996)		(*n* = 2653)		(*n* = 4338)		(*n* = 4249)		(*n* = 4344)	
	Mean	(SD)	Mean	(SD)	Mean	(SD)	Mean	(SD)	Mean	(SD)	Mean	(SD)	Mean	(SD)	Mean	(SD)
Foods intake																
Cereals (g)	509	(199)	523	(192)	532	(191)	550	(195)	352	(143)	364	(134)	380	(135)	390	(140)
Potatoes and starches (g)	48.4	(66.3)	53.3	(66.6)	58.4	(72.0)	65.2	(81.0)	47.7	(63.4)	49.6	(60.1)	54.4	(64.3)	62.9	(72.3)
Sugars and sweeteners (g)	4.6	(9.3)	5.8	(9.2)	7.3	(8.8)	9.3	(10.3)	4.6	(7.7)	6.1	(8.3)	7.2	(8.4)	9.3	(9.7)
Pulses (g)	61.5	(99.5)	66.8	(81.5)	73.1	(83.3)	71.9	(85.7)	56.3	(77.6)	62.3	(72.3)	65.5	(72.0)	69.3	(78.5)
Nuts and seeds (g)	2.8	(12.7)	2.8	(10.2)	2.6	(8.8)	2.7	(9.6)	2.6	(9.5)	2.8	(9.0)	2.9	(9.4)	2.8	(9.8)
Vegetables (g)	280	(195.4)	298	(176.4)	311	(185.9)	309	(191.6)	262	(167.6)	283	(161.5)	301	(168.5)	301	(175.6)
Fruits (g)	98	(151)	101	(137)	108	(158)	105	(145)	111	(135)	117	(130)	129	(138)	131	(145)
Mushrooms (g)	18.4	(34.9)	18.0	(30.8)	19.0	(32.9)	19.3	(31.7)	16.5	(27.6)	17.6	(26.1)	18.4	(28.2)	19.6	(29.0)
Algae (g)	9.6	(22.7)	12.5	(24.2)	12.8	(22.8)	11.7	(23.9)	7.2	(16.1)	11.0	(25.9)	12.1	(21.5)	11.7	(25.0)
Fish and shellfish (g)	61.5	(69.3)	80.2	(73.7)	94.2	(83.5)	90.8	(85.3)	48.9	(57.2)	63.9	(58.3)	76.6	(67.4)	75.5	(70.3)
Meats (g)	95.6	(81.5)	95.3	(75.6)	98.8	(79.8)	106.6	(84.9)	72.0	(58.6)	74.2	(58.3)	73.7	(59.5)	80.5	(66.0)
Eggs (g)	33.0	(36.4)	39.9	(37.2)	39.0	(35.9)	38.9	(38.6)	28.0	(31.7)	35.2	(31.6)	34.7	(32.7)	33.8	(33.3)
Milks (g)	110	(152)	108	(141)	106	(144)	90	(125)	123	(142)	126	(133)	120	(131)	113	(129)
Fats and oils (g)	10.9	(10.4)	11.8	(10.1)	11.7	(10.2)	11.5	(10.4)	8.8	(8.5)	9.6	(8.5)	9.9	(8.9)	9.5	(9.1)
Confectioneries (g)	25.8	(50.2)	23.6	(46.2)	21.7	(41.9)	23.7	(45.9)	30.8	(48.9)	30.2	(44.9)	30.8	(48.7)	31.0	(46.7)
Beverages (g)	781	(551)	863	(622)	871	(594)	845	(575)	672	(458)	687	(445)	671	(441)	693	(425)
Seasonings and spices (g)	84.5	(90.2)	83.3	(78.0)	101.1	(91.0)	134.0	(119.2)	67.6	(66.1)	70.3	(67.8)	83.7	(77.3)	107.7	(93.4)
Nutrients intake																
Energy (kcal)	2096	(606)	2129	(547)	2195	(541)	2251	(593)	1633	(461)	1689	(420)	1763	(437)	1806	(451)
Protein (g)	70.2	(24.6)	74.6	(22.4)	79.5	(23.0)	80.9	(24.2)	58.2	(19.6)	63.0	(18.7)	66.7	(19.8)	68.8	(20.5)
Protein (%energy)	13.5	(3.1)	14.1	(2.9)	14.6	(3.0)	14.5	(2.9)	14.3	(3.4)	15.0	(3.0)	15.3	(3.0)	15.3	(3.0)
Fat (%energy)	25.7	(7.8)	24.6	(7.0)	24.6	(7.2)	24.4	(7.2)	27.7	(7.9)	27.3	(7.3)	26.6	(7.2)	26.4	(7.3)
Saturated fat (%energy)	7.11	(2.82)	6.46	(2.39)	6.33	(2.40)	6.33	(2.40)	7.88	(3.07)	7.47	(2.70)	7.11	(2.61)	7.04	(2.62)
*n*-3 PUFA (%energy)	0.90	(0.60)	1.04	(0.57)	1.11	(0.60)	1.03	(0.61)	0.93	(0.62)	1.07	(0.59)	1.13	(0.62)	1.10	(0.64)
*n*-6 PUFA (%energy)	4.25	(1.82)	4.40	(1.65)	4.42	(1.71)	4.30	(1.77)	4.45	(1.90)	4.68	(1.77)	4.67	(1.77)	4.60	(1.82)
Carbohydrate (%energy)	56.0	(9.8)	55.9	(9.2)	55.3	(9.2)	55.4	(9.4)	56.1	(9.5)	55.7	(8.6)	56.1	(8.5)	56.1	(8.6)
Dietary fiber (g)	14.9	(7.0)	15.5	(6.8)	16.2	(7.2)	16.3	(7.2)	13.8	(6.3)	14.6	(6.3)	15.6	(6.6)	16.0	(7.1)
Vitamin A (μg RE)	478	(610)	546	(600)	606	(862)	587	(1050)	448	(437)	515	(465)	574	(644)	553	(747)
Thiamine (mg)	0.98	(0.63)	0.93	(0.59)	0.98	(0.58)	0.99	(0.62)	0.79	(0.42)	0.80	(0.38)	0.83	(0.45)	0.85	(0.45)
Riboflavin (mg)	1.23	(0.69)	1.25	(0.65)	1.31	(0.69)	1.29	(0.70)	1.06	(0.54)	1.13	(0.50)	1.18	(0.54)	1.18	(0.54)
Niacin (NE mg)	14.9	(7.1)	16.4	(7.7)	17.9	(8.1)	18.2	(8.7)	12.4	(5.6)	13.5	(5.8)	14.6	(6.3)	15.1	(6.6)
Vitamin B_6_ (mg)	1.15	(0.64)	1.26	(0.81)	1.36	(0.88)	1.38	(0.85)	0.98	(0.53)	1.06	(0.49)	1.15	(0.59)	1.18	(0.55)
Vitamin B_12_ (μg)	5.5	(5.9)	6.7	(6.9)	7.9	(8.3)	7.4	(8.1)	4.4	(4.9)	5.5	(5.5)	6.4	(6.5)	6.1	(6.6)
Folate (μg)	283	(142)	312	(146)	333	(175)	330	(186)	263	(129)	292	(130)	314	(149)	314	(148)
Vitamin C (mg)	92	(78)	100	(77)	106	(86)	104	(81)	94	(75)	103	(75)	111	(80)	111	(83)
Na (mg)	3901	(1647)	3956	(1488)	4503	(1554)	5099	(1773)	3191	(1298)	3360	(1229)	3844	(1303)	4446	(1544)
K (mg)	2241	(954)	2413	(899)	2574	(951)	2576	(996)	2065	(859)	2232	(831)	2398	(887)	2465	(931)
Ca (mg)	513	(290)	521	(261)	543	(273)	531	(262)	478	(260)	506	(250)	529	(256)	530	(260)
Mg (mg)	244	(95)	264	(94)	284	(96)	289	(102)	214	(84)	234	(83)	252	(86)	262	(94)
Fe (mg)	7.5	(3.2)	8.1	(3.0)	8.7	(3.2)	8.9	(3.5)	6.6	(2.7)	7.3	(2.9)	7.9	(2.9)	8.2	(3.2)
Zn (mg)	8.4	(3.2)	8.8	(2.8)	9.2	(2.9)	9.4	(3.0)	6.8	(2.4)	7.3	(2.2)	7.6	(2.4)	7.8	(2.4)
Cu (mg)	1.21	(0.45)	1.28	(0.42)	1.34	(0.52)	1.33	(0.48)	1.00	(0.38)	1.07	(0.35)	1.13	(0.37)	1.14	(0.39)

^a^*p* for trend used in the general linear model were all <0.001. PUFA, polyunsaturated fatty acid; SD, standard deviation.

**Table 4 nutrients-10-01865-t004:** Characteristics of the dietary intakes of the participants according to the portion size of miso.

	Men								Women							
	Group 1 (0 g)		Group 2 (0–<9.0 g)		Group 3 (9.0–<13.3 g)		Group 4 (≥13.3 g) ^a^		Group 1 (0 g)		Group 2 (0–<8.67 g)		Group 3 (8.67–<12.0 g)		Group 4 (≥12.0 g) ^a^	
	(*n* = 3615)		(*n* = 2410)		(*n* = 1932)		(*n* = 2197)		(*n* = 5404)		(*n* = 3365)		(*n* = 3384)		(*n* = 3431)	
	Mean	(SD)	Mean	(SD)	Mean	(SD)	Mean	(SD)	Mean	(SD)	Mean	(SD)	Mean	(SD)	Mean	(SD)
Foods intake																
Cereals (g)	514	(201)	525	(183)	550	(193)	552	(191)	359	(140)	370	(135)	382	(134)	391	(140)
Potatoes and starches (g)	49.0	(66.3)	58.8	(69.6)	63.2	(77.6)	65.4	(80.4)	47.2	(62.2)	57.0	(63.8)	58.4	(67.6)	58.8	(69.7)
Sugars and sweeteners (g)	6.7	(9.7)	7.5	(10.4)	7.2	(8.7)	7.4	(9.2)	6.5	(8.5)	7.2	(8.9)	7.0	(8.4)	7.8	(9.6)
Pulses (g)	53.1	(88.0)	76.0	(79.6)	78.3	(81.4)	81.1	(88.9)	49.9	(75.1)	71.1	(74.3)	68.9	(71.6)	74.8	(75.7)
Nuts and seeds (g)	2.6	(10.9)	2.8	(8.5)	2.8	(9.7)	2.7	(10.3)	2.7	(9.6)	2.9	(8.2)	3.1	(11.1)	2.8	(8.4)
Vegetables (g)	265	(187.1)	319	(181.8)	318	(178.0)	333	(187.7)	253	(162.2)	306	(168.7)	304	(166.5)	317	(172.6)
Fruits (g)	91	(142)	111	(155)	108	(149)	112	(145)	109	(134)	127	(132)	127	(138)	137	(147)
Mushrooms (g)	15.2	(29.9)	20.0	(31.1)	19.2	(29.0)	22.7	(38.7)	15.7	(27.4)	18.9	(27.1)	18.7	(27.1)	21.0	(29.4)
Algae (g)	7.6	(20.1)	14.5	(26.9)	14.1	(22.4)	14.3	(24.7)	6.8	(18.1)	12.7	(29.4)	12.8	(21.9)	13.4	(23.4)
Fish and shellfish (g)	70.0	(75.8)	92.9	(76.5)	89.5	(79.9)	96.0	(87.6)	55.5	(62.9)	73.2	(63.9)	74.2	(63.6)	76.6	(67.8)
Meats (g)	107.3	(84.8)	91.3	(72.6)	98.1	(79.4)	97.7	(81.3)	79.1	(63.8)	71.5	(57.4)	73.9	(59.3)	75.3	(61.2)
Eggs (g)	37.9	(38.4)	37.9	(35.4)	39.1	(36.7)	39.3	(37.5)	32.8	(33.9)	32.6	(30.7)	33.8	(31.9)	34.9	(32.5)
Milks (g)	103	(145)	107	(136)	100	(136)	97	(134)	121	(139)	124	(131)	116	(126)	121	(134)
Fats and oils (g)	12.5	(10.7)	10.8	(9.8)	11.2	(9.9)	11.2	(10.4)	10.1	(9.2)	8.7	(8.3)	9.4	(8.3)	9.6	(8.9)
Confectioneries (g)	25.0	(48.9)	23.7	(45.2)	21.4	(42.4)	22.3	(42.5)	33.3	(49.3)	29.3	(44.7)	28.6	(46.0)	30.0	(47.0)
Beverages (g)	875	(640)	818	(536)	837	(597)	846	(561)	709	(460)	657	(432)	659	(418)	684	(437)
Seasonings and spices (g)	88.6	(93.3)	98.9	(94.8)	113.3	(99.5)	123.8	(107.7)	71.3	(72.5)	80.1	(74.5)	91.6	(85.0)	100.2	(85.8)
Nutrients intake																
Energy (kcal)	2124	(598)	2144	(527)	2225	(570)	2266	(554)	1675	(457)	1709	(420)	1748	(431)	1828	(448)
Protein (g)	72.6	(23.7)	77.4	(22.2)	79.6	(23.7)	82.7	(23.8)	60.2	(20.0)	65.2	(19.2)	66.6	(19.1)	70.1	(20.1)
Protein (%energy)	13.8	(3.1)	14.5	(2.8)	14.4	(2.9)	14.7	(3.0)	14.4	(3.2)	15.3	(3.0)	15.3	(2.9)	15.4	(3.0)
Fat (%energy)	25.8	(7.7)	24.1	(7.1)	24.1	(6.8)	24.0	(6.9)	27.9	(7.7)	26.4	(7.4)	26.3	(7.1)	26.5	(7.1)
Saturated fat (%energy)	6.99	(2.67)	6.24	(2.36)	6.15	(2.25)	6.17	(2.30)	7.89	(2.95)	7.12	(2.64)	6.94	(2.51)	6.98	(2.54)
*n*-3 PUFA (%energy)	2.19	(1.54)	2.60	(1.55)	2.71	(1.68)	2.79	(1.66)	1.76	(1.26)	2.15	(1.33)	2.24	(1.35)	2.36	(1.41)
*n*-6 PUFA (%energy)	10.26	(5.26)	10.30	(4.90)	10.93	(5.06)	11.13	(5.02)	8.44	(4.35)	8.77	(4.23)	9.13	(4.23)	9.73	(4.40)
Carbohydrate (%energy)	54.8	(9.9)	56.2	(9.1)	56.1	(9.0)	55.9	(9.0)	55.3	(9.1)	56.3	(8.7)	56.5	(8.4)	56.2	(8.4)
Dietary fiber (g)	13.8	(6.5)	16.4	(7.0)	16.8	(7.2)	17.8	(7.2)	13.2	(6.0)	15.6	(6.8)	15.9	(6.6)	16.9	(6.8)
Vitamin A (μg RE)	530	(827)	597	(960)	577	(752)	580	(741)	486	(509)	555	(638)	541	(603)	565	(691)
Thiamine (mg)	0.95	(0.71)	0.96	(0.53)	0.97	(0.50)	1.01	(0.57)	0.78	(0.44)	0.81	(0.34)	0.83	(0.39)	0.88	(0.49)
Riboflavin (mg)	1.23	(0.76)	1.29	(0.64)	1.29	(0.63)	1.33	(0.64)	1.09	(0.55)	1.14	(0.48)	1.15	(0.50)	1.22	(0.57)
Niacin (NEmg)	16.0	(8.3)	17.3	(7.7)	17.6	(8.0)	18.3	(8.1)	13.0	(6.0)	14.2	(6.1)	14.5	(6.2)	15.1	(6.4)
Vitamin B_6_ (mg)	1.19	(0.82)	1.35	(0.79)	1.37	(0.88)	1.40	(0.81)	1.00	(0.55)	1.13	(0.53)	1.15	(0.51)	1.20	(0.57)
Vitamin B_12_ (μg)	5.8	(6.8)	7.8	(8.2)	7.4	(7.2)	8.0	(8.3)	4.6	(5.1)	6.3	(6.3)	6.2	(6.4)	6.4	(6.5)
Folate (μg)	280	(157)	334	(178)	338	(160)	352	(164)	261	(128)	310	(139)	315	(141)	333	(152)
Vitamin C (mg)	88	(76)	110	(83)	109	(86)	110	(78)	92	(75)	111	(78)	111	(79)	118	(82)
Na (mg)	4004	(1634)	4225	(1527)	4620	(1595)	5228	(1719)	3306	(1334)	3566	(1279)	3954	(1362)	4502	(1485)
K (mg)	2228	(924)	2564	(931)	2605	(945)	2707	(959)	2072	(835)	2379	(876)	2409	(878)	2536	(919)
Ca (mg)	485	(273)	546	(262)	544	(260)	570	(268)	469	(256)	534	(261)	522	(247)	559	(254)
Mg (mg)	245	(96)	281	(93)	289	(96)	302	(98)	215	(83)	250	(88)	254	(85)	270	(89)
Fe (mg)	7.4	(3.2)	8.6	(3.1)	8.9	(3.2)	9.5	(3.3)	6.6	(2.7)	7.8	(3.2)	8.0	(2.8)	8.6	(3.0)
Zn (mg)	8.5	(3.1)	9.0	(2.7)	9.4	(2.8)	9.6	(3.0)	7.0	(2.5)	7.4	(2.2)	7.6	(2.3)	8.0	(2.4)
Cu (mg)	1.16	(0.42)	1.34	(0.52)	1.38	(0.47)	1.44	(0.44)	0.97	(0.37)	1.12	(0.35)	1.16	(0.36)	1.22	(0.38)

^a^*p* for trend used in the general linear model were all <0.001. PUFA, polyunsaturated fatty acid; SD, standard deviation.

**Table 5 nutrients-10-01865-t005:** Association between the portion sizes of traditional Japanese seasonings and blood pressure.

		Crude model		Model 1 ^a^		Model 2 ^b^	
		Lsmean	(95% CI)	Lsmean	(95% CI)	Lsmean	(95% CI)
**Soy sauce**							
Men (*n* = 10,154)							
SBP	Group 1 (0 g)	131.2	(130.3, 132.1)	131.9	(131.0, 132.7)	129.9	(126.5, 133.2)
	Group 2 (0–<4.0 g)	131.7	(131.1, 132.4)	131.8	(131.2, 132.4)	129.9	(126.6, 133.1)
	Group 3 (4.0–<7.23 g)	132.7	(132.0, 133.3)	132.2	(131.6, 132.8)	130.1	(126.8, 133.4)
	Group 4 (≥7.23 g)	131.9	(131.3, 132.6)	132.0	(131.4, 132.6)	129.7	(126.4, 133.0)
	*p* for trend		0.231		0.328		0.728
DBP	Group 1 (0 g)	81.7	(81.1, 82.3)	81.8	(81.2, 82.4)	80.3	(78.0, 82.6)
	Group 2 (0–<4.0 g)	81.4	(81.0, 81.8)	81.4	(81.0, 81.8)	79.8	(77.6, 82.1)
	Group 3 (4.0–<7.23 g)	81.6	(81.2, 82.1)	81.6	(81.2, 82.0)	80.0	(77.7, 82.2)
	Group 4 (≥7.23 g)	81.3	(80.9, 81.7)	81.3	(80.9, 81.7)	79.5	(77.3, 81.8)
	*p* for trend		0.739		0.694		0.262
Women (*n* = 15,584)							
SBP	Group 1 (0 g)	123.1	(122.4, 123.8)	124.7	(124.1, 125.3)	126.2	(123.6, 128.9)
	Group 2 (0–<3.5 g)	124.2	(123.6, 124.8)	124.4	(123.9, 124.9)	126.1	(123.5, 128.7)
	Group 3 (3.5–<6.5 g)	125.2	(124.6, 125.7)	124.6	(124.1, 125.1)	126.1	(123.5, 128.7)
	Group 4 (≥6.5 g)	125.9	(125.4, 126.5)	125.3	(124.8, 125.8)	126.7	(124.1, 129.3)
	*p* for trend		<0.001		0.026		0.108
DBP	Group 1 (0 g)	76.0	(75.6, 76.4)	76.5	(76.1, 76.9)	75.9	(74.2, 77.6)
	Group 2 (0–<3.5 g)	76.3	(75.9, 76.6)	76.3	(76.0, 76.6)	75.9	(74.2, 77.5)
	Group 3 (3.5–<6.5 g)	76.3	(76.0, 76.7)	76.2	(75.9, 76.5)	75.6	(73.9, 77.3)
	Group 4 (≥6.5 g)	76.6	(76.2, 76.9)	76.4	(76.1, 76.7)	75.6	(74.0, 77.3)
	*p* for trend		0.190		0.689		0.158
**Miso**							
Men (*n* = 10,154)							
SBP	Group 1 (0 g)	131.3	(130.8, 131.9)	132.6	(132.1, 133.2)	130.4	(127.1, 133.7)
	Group 2 (0–<9.0 g)	132.2	(131.5, 132.9)	131.2	(130.6, 131.9)	129.3	(126.0, 132.6)
	Group 3 (9.0–<13.3 g)	132.9	(132.1, 133.6)	132.4	(131.6, 133.1)	130.2	(126.9, 133.5)
	Group 4 (≥13.3 g)	132.1	(131.3, 132.8)	131.4	(130.8, 132.1)	129.2	(125.9, 132.5)
	*p* for trend		0.016		0.140		0.139
DBP	Group 1 (0 g)	81.7	(81.3, 82.1)	81.9	(81.5, 82.3)	80.2	(77.9, 82.4)
	Group 2 (0–<9.0 g)	81.0	(80.6, 81.5)	80.9	(80.4, 81.3)	79.3	(77.1, 81.6)
	Group 3 (9.0–<13.3 g)	81.6	(81.1, 82.1)	81.5	(81.0, 82.0)	79.8	(77.5, 82.1)
	Group 4 (≥13.3 g)	81.4	(81.0, 81.9)	81.3	(80.9, 81.8)	79.6	(77.3, 81.8)
	*p* for trend		0.938		0.379		0.278
Women (*n* = 15,584)							
SBP	Group 1 (0 g)	123.1	(122.6, 123.6)	125.1	(124.7, 125.5)	126.6	(124.0, 129.2)
	Group 2 (0–<8.67 g)	125.1	(124.5, 125.7)	124.0	(123.5, 124.6)	125.7	(123.0, 128.3)
	Group 3 (8.67–<12.0 g)	125.3	(124.7, 125.9)	124.4	(123.9, 125.0)	126.0	(123.4, 128.6)
	Group 4 (≥12.0 g)	126.6	(125.9, 127.2)	125.3	(124.8, 125.8)	126.7	(124.1, 129.3)
	*p* for trend		<0.001		0.415		0.566
DBP	Group 1 (0 g)	75.9	(75.6, 76.2)	76.5	(76.2, 76.8)	75.9	(74.3, 77.6)
	Group 2 (0–<8.67 g)	76.2	(75.9, 76.6)	75.9	(75.5, 76.3)	75.4	(73.7, 77.1)
	Group 3 (8.67–<12.0 g)	76.3	(75.9, 76.7)	76.0	(75.7, 76.4)	75.5	(73.8, 77.2)
	Group 4 (≥12.0 g)	77.1	(76.7, 77.5)	76.7	(76.4, 77.1)	76.0	(74.4, 77.7)
	*p* for trend		<0.001		0.553		0.804

^a^ Adjusted for age. ^b^ Adjusted for age, BMI, smoking status, amount of alcoholic beverages intake, number of steps, and vegetable and fruit intake. CI, confidence interval; DBP, diastolic blood pressure; SBP, systolic blood pressure; BMI, body mass index.

**Table 6 nutrients-10-01865-t006:** Odds ratio of the association between the portion sizes of traditional Japanese seasonings and hypertension.

	No. of Participants Cases/Total	Crude model		Model 1 ^a^		Model 2 ^b^	
		OR	(95% CI)	OR	(95% CI)	OR	(95% CI)
**Soy sauce**							
Men							
Q1 (0 g)	472/1350	1.00 (Ref.)		1.00 (Ref.)		1.00 (Ref.)	
Q2 (0–<4.0 g)	1092/2982	1.08	(0.94, 1.23)	1.03	(0.89, 1.18)	1.03	(0.89, 1.18)
Q3 (4.0–<7.23 g)	1078/2826	1.15	(1.00, 1.31)	1.05	(0.92, 1.21)	1.04	(0.90, 1.20)
Q4 (≥7.23 g)	1069/2996	1.03	(0.90, 1.18)	0.98	(0.85, 1.12)	0.94	(0.82, 1.09)
*p* for trend			0.368		0.459		0.865
Women							
Q1 (0 g)	547/2653	1.00 (Ref.)		1.00 (Ref.)		1.00 (Ref.)	
Q2 (0–<3.5 g)	962/4338	1.10	(0.98, 1.24)	0.96	(0.85, 1.09)	0.98	(0.86, 1.12)
Q3 (3.5–<6.5 g)	993/4249	1.17	(1.04, 1.32)	0.97	(0.86, 1.10)	0.97	(0.85, 1.11)
Q4 (≥6.5 g)	1066/4344	1.25	(1.11, 1.41)	1.05	(0.93, 1.19)	1.03	(0.91, 1.17)
*p* for trend			<0.001		0.108		0.237
**Miso**							
Men							
Q1 (0 g)	1299/3615	1.00 (Ref.)		1.00 (Ref.)		1.00 (Ref.)	
Q2 (0–<9.0 g)	873/2410	1.01	(0.91, 1.13)	0.84	(0.75, 0.94)	0.86	(0.77, 0.97)
Q3 (9.0–<13.3 g)	726/1932	1.07	(0.96, 1.20)	0.93	(0.83, 1.05)	0.93	(0.83, 1.05)
Q4 (≥13.3 g)	813/2197	1.05	(0.94, 1.17)	0.89	(0.80, 1.00)	0.89	(0.79, 1.00)
*p* for trend			0.144		0.292		0.270
Women							
Q1 (0 g)	1089/5404	1.00 (Ref.)		1.00 (Ref.)		1.00 (Ref.)	
Q2 (0–<8.67 g)	814/3365	1.26	(1.14, 1.40)	0.96	(0.86, 1.07)	0.97	(0.87, 1.09)
Q3 (8.67–<12.0 g)	785/3384	1.20	(1.08, 1.33)	0.93	(0.83, 1.03)	0.94	(0.84, 1.05)
Q4 (≥12.0 g)	880/3431	1.37	(1.24, 1.51)	1.04	(0.94, 1.16)	1.03	(0.93, 1.16)
*p* for trend			<0.001		0.312		0.443

^a^ Adjusted for age. ^b^ Adjusted for age, BMI, smoking status, amount of alcoholic beverages intake, number of steps, and vegetable and fruit intake. OR, odds ratio; CI, confidence interval; BMI, body mass index.

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
