# Peer review of "Association between the Portion Sizes of Traditional Japanese Seasonings—Soy Sauce and Miso—and Blood Pressure: Cross-Sectional Study Using National Health and Nutrition Survey, 2012–2016 Data"

_nutrients, 2018, doi:10.3390/nu10121865_

Reviewer 1 Report

The paper by Okada et al. is a well-written one. In their work about a very large data set from 2012-2016 NHNS, the authors address an interesting issue: may the high-salt Japanese seasoning be related to blood pressure?

Their answer seems “no”.

I find a number of problems and pitfalls to deal with in the analysis of the results as well as in the discussion.

First of all, the key take-home message I find in this paper is: blood pressure seems not related to salt intake; this is to be avoided, because it is not true.

2nd: we are now in the 21st century and blood pressure recording based on two single measures of BP by mercury sphygmomanometers is difficult to accept. In the methods section a number of things should be specified: the arm-related cuff employed, the position of the subject, the calibration of devices, the reproducibility of the method, even the brand employed should conform to validation procedures (it is enough to consult www.dableducational.org and the American Heart Association website).

3rd: in the analysis of the effects of the seasoning portions on BP, the authors divide the population under study in four different groups based on the referred consumption of miso or soy sauce. It seems possible that the groups are unbalanced/unmatched by BMI and alcohol status and I cannot find reference to this in the text. This has to be clarified.

4th: data are based on dietary recordings, and this (food frequency questionnaires, 24-hour recordings and so on) is the weakest way to assess sodium and potassium consumption. This is to be stressed.

5th: the possibility that sodium intake is misrepresented by the dietary questionnaires must be taken into account.

6th: in the population under exam the consumption of sodium is very high. For instance, already in the first group (those with the lowest seasoning consumption) the reported sodium intake is 8 g/d for women and 10 g/d for men. The salt consumption progressively increases in the following groups, without relationship with the values of BP (see point 1). Potassium intake seems quite low (I find a Na/K ratio near 2 for men and near 1.5 in women, quite far from the ideal values).

7th: I have trouble understanding how anyone can calculate so sharp data for sodium and potassium intakes from dietary recordings.

8th: in the table 2 on page 6 of the manuscript “Demographic characteristics of the participants according to the portion size of miso”, the total numbers of the four male groups are wrong: the authors report 2653; 4338; 4249; 4344 – instead of 3615; 2410; 1932; 2197 (that seem the correct ones), it is amazing that the sum of the wrong numbers is 15584, i.e. the total number for the women groups, perhaps some cut & paste mistake? Or the numbers are inverted with the women groups in table 1 (see page 5)?

9th: the same mistakes are present in the tables 3 and 4.

10th: in the discussion (rows 205-206) the authors state that in 4 grams of soy sauce there are 228 grams of sodium and in 9 grams of miso there are 441 grams of sodium; I believe there is something wrong… ”Soy sauce (4 g) contains 228 g of sodium (koikuchi-shoyu) and miso (9.0 g) contains 441 g of sodium (rice-koji miso)”.

11th: the authors cite in the introduction the reference #5 (page 1, rows 40,41) when they say: “A previous study observed a positive relationship between dietary salt intake and BP levels”. Actually I would cite a very high number of papers instead. May be the authors intended “in the Japanese population”? This should be stated.

12th: why a previous paper reporting contrasting results has not been cited? Circ. J. 2003; 67: 530-4. Feasibility and effect on blood pressure of 6-week trial of low sodium soy sauce and miso (fermented soybean paste). Nakamura M, Aoki N, Yamada T, Kubo N. This would add interest to the discussion.

Reviewer 2 Report

General Comment:  The study investigated the association of portion sizes of soy sauce and miso on blood pressure in Japanese registered in the 2012-2016 NHNS in a cross-sectional study.  After a thorough analysis, authors concluded no association exists between portion sizes and blood pressure.  This study is interesting and important as well as impressive.  The paper is well written and understandable.  All sections, Introduction, Methods, Results, and Discussion, seem very organized. I have a few suggestions for the authors to consider.

Guidelines for hypertension were revised by the American Heart Association in 2017 and the >140/90 value is now considered Stage 2 hypertension, with Stage 1 being 130-139/80-89, and Elevated Hypertension being 120-129/80-89.  Although it is not critical for this paper, it might be appropriate just to acknowledge the AHA revisions and state if would impact the results of this study. 

There were 7,279 hypertensives (line 173) in this study of 25,738 participants, which is about 28%.  Does this not indicate that the sample used in this study is not representative of the Japanese population with 8% hypertensives (line 37)?

I commend the authors for their use of three statistical models of analysis and I agree with the overall conclusion that portion sizes of these seasonings are not associated with blood pressure.  The crude model did show significance in women but not the age-adjusted or multivariate models.  Evidence suggests that women can become salt-sensitive upon menopause.  Could this be an explanation for the Crude-model results?  Can the number of premenopausal and postmenopausal women be determined or estimated?

Japanese have less hypertension compared to most of rest of the industrial world yet they consume higher salt (line 40).  Might the consumption of the two seasonings have something to do with that observation and be related, at least indirectly, to the statement made in the Conclusion (lines 266-268)? 

Lines 205 & 206:  Should that be 228 mg and 441 mg rather than g?

Line 264: Suggest adding the phrase “…when adjusting for appropriate risk factors….”; after “In conclusion,” 

 Author Response

Round  2

Reviewer 1 Report

The authors modified the text as suggested and corrected the few mistakes in the tables. I appreciated very much their response to my previous comment "1", where they say: "Participants  with a larger portion size of soy sauce or miso had a higher  consumption of vegetables, fruits, fish, and soy products in the present  study. Some cohort studies in Japan have  shown that the dietary pattern characterized by high consumption  of vegetables, fish, soy products and seaweed, were associated with a  reduced risk of CVD (Shimazu  et al, 2007; Maruyama et al, 2013). Therefore, the larger portion size  of soy sauce or miso may not have been associated with hypertension  according to traditional Japanese dietary patterns in our study".

Thereafter I  suggest the authors to paste this sentence near the end of the discussion, where they deemed appropriate.

Author Response

Reviewer #1

Thank you for your review. We have revised the manuscript based on your comments. We have indicated the page and line numbers referenced in response to your comments. All revised sentences are highlighted in grey in our revised manuscript, as well as in this response letter.

 Referee: 1

Comments to the Author

The authors modified the text as suggested and corrected the few mistakes in the tables. I appreciated very much their response to my previous comment "1", where they say: "Participants with a larger portion size of soy sauce or miso had a higher consumption of vegetables, fruits, fish, and soy products in the present study. Some cohort studies in Japan have shown that the dietary pattern characterized by high consumption of vegetables, fish, soy products and seaweed, were associated with a reduced risk of CVD (Shimazu et al, 2007; Maruyama et al, 2013). Therefore, the larger portion size of soy sauce or miso may not have been associated with hypertension according to traditional Japanese dietary patterns in our study".

Thereafter I suggest the authors to paste this sentence near the end of the discussion, where they deemed appropriate.

 Response: Thank you for your suggestion. We have added the following text and reference to the manuscript:

“The demographic and dietary characteristics of participants with larger portion sizes of soy sauce or miso correspond to older people and those with a higher intake of almost all nutrients and foods including vegetables, fruits, fish, and soy products; this may offset the effects of soy sauce or miso on BP. Some cohort studies in Japan have shown that the dietary pattern characterized by high consumption of vegetables, fish, soy products and seaweed, were associated with a reduced risk of CVD [23, 24]. Therefore, the larger portion size of soy sauce or miso may not have been associated with hypertension according to traditional Japanese dietary patterns in our study. (p. 9–p. 10, line 243–250)

23.       Maruyama, K.; Iso, H, Date.; C, Kikuchi, S.; Watanabe, Y.; Wada, Y.; Inaba, Y.; Tamakoshi, A.; JACC Study Group. Dietary patterns and risk of cardiovascular deaths among middle-aged Japanese: JACC Study. Nutr Metab Cardiovasc Dis. 2013, 23, 519-527. doi: 10.1016/j.numecd.2011.10.007

24.         Shimazu, T.; Kuriyama, S.; Hozawa, A.; Ohmori, K.; Sato, Y.; Nakaya, N.; Nishino, Y.; Tsubono, Y.; Tsuji, I. Dietary patterns and cardiovascular disease mortality in Japan: a prospective cohort study. Int J Epidemiol. 2007, 36, 600-609. doi: 10.1093/ije/dym005” (p. 15, line 386–392)

Reviewer 2 Report

The authors have responded to all my concerns.  I thank them for responding in detail to the issues I raised. 

Round  3

Reviewer 1 Report

I have no additional comments.